# Osteoprotegerin Is a Better Predictor for Cardiovascular and All-Cause Mortality than Vascular Calcifications in a Multicenter Cohort of Patients on Peritoneal Dialysis

**DOI:** 10.3390/biom12040551

**Published:** 2022-04-08

**Authors:** Marcela Ávila, Ma. del Carmen Prado, Renata Romero, Ricardo Córdova, Ma. del Carmen Rigo, Miguel Trejo, Carmen Mora, Ramón Paniagua

**Affiliations:** 1Unidad de Investigación Médica en Enfermedades Nefrológicas, Hospital de Especialidades, CMN SXXI, Instituto Mexicano del Seguro Social, Ciudad de México 06720, Mexico; carpradou@gmail.com (M.d.C.P.); renata.romeros@gmail.com (R.R.); mrigo_islas@hotmail.com (M.d.C.R.); miguel.trejo.villeda@gmail.com (M.T.); moravillas@gmail.com (C.M.); jrpaniaguas@gmail.com (R.P.); 2Departamento de Radiología e Imagen, Hospital de Especialidades, CMN SXXI, Instituto Mexicano del Seguro Social, Ciudad de México 06720, Mexico; ric.cordova.rx@gmail.com

**Keywords:** vascular calcification, diabetes mellitus, osteoprotegerin, cardiovascular mortality, risk factor, peritoneal dialysis

## Abstract

The purpose of this study was to compare vascular calcification (VC), serum osteoprotegerin (OPG) levels, and other biochemical markers to determine their value as available predictors of all-cause and cardiovascular (CV) mortality in patients on peritoneal dialysis (PD). A total of 197 patients were recruited from seven dialysis centers in Mexico City. VC was assessed with multi-slice computed tomography, measured using the calcification score (CaSc). OPG, albumin, calcium, hsC-reactive protein, phosphorous, osteocalcin, total alkaline phosphatase, and intact parathormone were also analyzed. Follow-up and mortality analyses were assessed using the Cox regression model. The mean age was 43.9 ± 12.9 years, 64% were males, and 53% were diabetics. The median OPG was 11.28 (IQR: 7.6–17.4 pmol/L), and 42% of cases had cardiovascular calcifications. The median VC was 424 (IQR:101–886). During follow-up (23 ± 7 months), there were 34 deaths, and 44% were cardiovascular in origin. In multivariable analysis, OPG was a significant predictor for all-cause (HR 1.08; *p* < 0.002) and CV mortality (HR 1.09; *p* < 0.013), and performed better than VC (HR 1.00; *p* < 0.62 for all-cause mortality and HR 1.00; *p* < 0.16 for CV mortality). For each mg/dL of albumin-corrected calcium, there was an increased risk for CV mortality, and each g/dL of albumin decreased the risk factor for all-cause mortality. OPG levels above 14.37 and 13.57 pmol/L showed the highest predictive value for all-cause and CV mortality in incident PD patients and performed better than VC.

## 1. Introduction

Cardiovascular diseases (CVD) are the main cause of comorbidity and mortality in the population with chronic kidney disease (CKD) [1], but the incidence has not been fully explained by traditional risk factors. Cardiovascular calcifications have been included among non-traditional risk factors; they may involve cardiac valves and the intima layer (atherosclerosis) or the middle layer (arteriosclerosis) of coronary arteries and peripheral vessels [2].

Arterial calcifications are a frequent finding in patients with CKD, even in non-dialysis patients, and they are directly associated with the extent of renal damage [3]. Moreover, vascular calcifications (VC) are an important marker of cardiovascular risk [4].

Various studies have shown that their presence is associated with greater all-cause and CV mortality in hemodialysis patients [5]. On the basis of this association, coronary artery calcification, quantified by means of multi-detector spiral computed tomography (CT), has been suggested as a screening test to assess cardiovascular risk in patients undergoing renal replacement therapy. However, the limited availability of the technique and its operator dependency complicate its routine use [6] and have prompted the need for other calcification biomarkers. Moreover, one of the more controversial points comes from controlled clinical trials, in which various interventions, aimed at modifying the evolution of coronary artery calcifications, showed little or no effect on mortality [7].

In recent years, various studies have highlighted the value of some bone metabolism-related proteins as biomarkers of vascular wall calcification. Among them, osteoprotegerin (OPG) seems to be particularly promising. OPG is a soluble glycoprotein, belonging to the soluble proteins of the tumor necrosis factor (TNF) receptor superfamily, and is classified as an osteoclastogenesis inhibition factor [8] because it is a decoy of the receptor activator of nuclear factor kappa-β ligand (RANKL) and TNF-related apoptosis-inducing ligand (TRAIL) [9]. OPG is expressed in most human tissues, including bone and vasculature (endothelial and vascular smooth muscle cells (VSMC)). It is induced by inflammatory cytokines, such as pro-inflammatory mediators, such as TNFα [10].

Although studies in vitro and in animal models suggest that OPG inhibits vascular calcification, clinical studies suggest that elevated serum OPG levels are directly associated with vascular calcifications, coronary artery disease, stroke, and future cardiovascular events [11], and common carotid artery intima-media thickness (CCA-IMT). Moreover, it has been shown to be a prognostic marker of cardiovascular risk in dialyzed patients [12] and of mortality in hemodialysis (HD) patients [13]. Furthermore, in diabetic patients without CKD, elevated circulating OPG levels are associated with acute myocardial infarction and chronic heart failure of ischemic etiology [14], and mortality in patients with angina pectoris [15]. Additionally, OPG is a predictor of mortality in patients with other diseases, such as cancer or amyloidosis [16,17].

These findings may suggest that OPG could serve to evaluate cardiovascular risk in dialysis patients. This may be of greater importance in younger patients without evident comorbidities, as, in this context, it may lead to identifying subjects that need a cardiovascular evaluation and specific interventions, and may support treatment modulation. 

The present study aimed to evaluate the role of osteoprotegerin compared to vascular calcification, and some mineral metabolism markers with all-cause and cardiovascular mortality in a multi-center cohort of patients on peritoneal dialysis (PD).

## 2. Materials and Methods

### 2.1. Study Design

A prospective observational cohort of incident patients in PD programs was recruited from 7 hospitals belonging to the National Network of the lnstituto Mexicano of Seguro Social (IMSS) in Mexico City and was followed up for at least 16 months. All-cause and CV mortality were the primary end points of the study.

### 2.2. Patient Selection

Participation was offered to all the patients who began peritoneal dialysis (>3 months and <4 months), either continuous ambulatory peritoneal dialysis (CAPD) or automated PD (APD) treatment, at the PD centers of each hospital. Written informed consent was obtained from the patients.

The inclusion criteria were: adult (>18 years), free of acute complications, including peritonitis and hospitalizations during the month prior to enrolment. Patients with previously known CVD (defined as heart failure, ischemic disease, arrhythmia, myocardial infarction)**,** chronic infections, malignancies, chronic obstructive pulmonary disease, ongoing steroid therapy, positivity for hepatitis B or C and HIV were excluded. Patients with incomplete data were also excluded.

### 2.3. Dialysis Schedule

Patients received 2 L, four times a day, for CAPD and four or five exchanges per night and a wet day for APD. Only dextrose solutions were available and the concentration was prescribed by the attending nephrologist according to the patient’s needs. Dialysis adequacy was calculated by total Kt/V: renal Kt/V+ peritoneal Kt/V and peritoneal ultrafiltration. Kt/V is a number used to quantify the adequacy of peritoneal dialysis and hemodialysis, and represents the clearance of Urea by the peritoneum and/or by the kidney, normalized by total body water. K is the clearance of urea by the peritoneum or the kidney, in ml/min, t is the time on dialysis (min), and V is the volume of distribution of urea, approximately equal to the volume of the patient’s total body water. The minimal recommended values were a weekly urea Kt/V of 1.7 and a daily ultrafiltration of 750 mL [18]. Residual renal function was calculated as the mean of urea and creatinine clearance.

### 2.4. Data Collection

Demographic and relevant clinical data were collected from medical records by trained nurses. A CT was performed at the baseline stage. Patients were censored at the end of the follow-up: kidney transplantation, or in the shift to hemodialysis or transfer to other hospitals, in case of voluntary withdrawal or death. Causes of death were obtained from the death certificate and were reviewed according to a caregiver.

### 2.5. Biochemical Assessments

Blood samples were drawn from an antecubital vein without stasis, after overnight fasting. The samples were centrifuged and the plasma and serum were separated and stored at –70 °C until assayed. Osteoprotegerin (OPG) was determined using ELISA (MicroVue Eia Kit. Quidel Corp. Specialty Products, San Diego, CA, USA). The intra-assay precision was 3% and the inter-assay precision was 4.5%, with a limit of detection of 1.16 to 60 pmol/L. N-MID osteocalcin and intact parathormone (iPTH) were analyzed by electrochemiluminescence immunoassay (Elecsys Modular Analytics 2010 Roche, Mannheim, Germany) The intra and inter coefficients of variation (% CoV) were 2.5% and 2.0%, respectively. Serum phosphorous (P), serum albumin (Alb), and albumin-corrected calcium (cCa) were calculated with the formula: cCa = (Ca (mg/dL) + 0.8(4-Alb g/dL)); total cholesterol (Chol), glucose (Glu), creatinine (Cr), total alkaline phosphatase (tALP), and high-sensitivity C-reactive protein (hsCRP) were measured using standard techniques (Hitachi 902 autoanalyser, Tokyo, Japan). Twenty-four-hour urine and dialysate collection was performed for both CAPD and APD patients.

### 2.6. Measurement of Arterial Calcifications

A 16-cut multi-slice computed tomography (MSCT) using Bright Speed, (GE, Beijing, China), was used to quantify the vascular calcifications (VC) in a standardized section of the abdominal aorta and pelvic vessels. The acquired images were reviewed using Advantage Workstation Swart Core software, v4.5, (GE, Waukesha, Wisconsin, WI, USA), and a calcification score (CaSc) was generated. The calcification score indicates the amount of calcified plaque in the arteries. The CaSc (AJ130 score) refers to the detection of densities of more than 130 Hounsfield units (HU) in areas of at least 1mm^2^. It is obtained from the product between the area of calcified plaque and its maximum density in Hounsfield units and is expressed in Agatston units (AU). A score of 0 implies the absence of calcified plaques, 1–10 AU shows minimal calcified plaques, 11–100 AU shows mild calcification, 101–300 AU shows moderate calcification, and >300 AU shows severe arterial calcification, as in other studies [19]. An example of a patient with calcifications is shown in Figure 1.

No contrast-enhancing agent was used; scanning was performed on sequential 2 mm thick layers. Two independent investigators, blind to the patient’s clinical history, evaluated the MSCT scans. Inter-observer reproducibility between the investigator and radiologist was assessed for all patients, with a coefficient of variation of 3.5%, the intra-observer coefficient of variation was 2.8% during three days, and the correlation between observers was 0.95 (CI%; 0.92 to 0.98). 

### 2.7. Statistical Analysis

Data are expressed as mean ± standard deviation (SD) in the case of continuous variables with normal distribution, or median and interquartile range (IQR) in case of non-normal distribution, or as frequencies in the case of categorical variables. Differences between groups were analyzed using a Student’s t-test, Mann–Whitney U test, or Chi-square test, as appropriate. Cox proportional hazards regression, using the Enter method, was used to estimate the all-cause and cardiovascular mortality hazard ratios, unadjusted and adjusted by: serum OPG, cCa, sAlb, P, iPTH, and hsCRP as predicting variables, and hsCRP was converted to a logarithm. The Cox models were determined with baseline co-variables. We additionally performed a Cox regression analysis using the forward conditional method to determine the score statistics for each variable.

Receiver operating characteristic (ROC) curves were made to determine the diagnostic OPG value. The area under curve (AUC) was evaluated, and we determined the concentration of OPG as the cut-off point for all-cause mortality and CV death, according to the Youden index method, with a significance of *p* < 0.05. Analysis was performed using SPSS Statistics for Windows, version 21.0. (IBM Corp, Armonk, NY, USA).

## 3. Results

Figure 1 shows the study flow chart: 328 patients were assessed for eligibility at the dialysis centers, and 230 of them were eligible. Thirty-three were excluded because of several causes: 18 patients did not meet the selection criteria, mainly because of age, catheter dysfunction, peritonitis, and hospitalizations during the previous month; 7 patients refused to participate; 4 patients moved to another city; 4 patients lost social security coverage. Thus, 197 patients were included in the final analysis.

### 3.1. Baseline Biochemical Data

Demographic and relevant clinical and biochemical data of the 197 included patients are shown inTable 1. The mean age was 43.9 ± 12.9 years old, 64% were males, and 53% were diabetics. The most frequent etiology was diabetic nephropathy, found in 52.8% of the patients. The median OPG was 11.28 pmol/L, with an interquartile range of 25–75; (7.6–17.1) comparable with other studies; 10.9 (IQR 8–10.3 pmol/L) [20].

A significant difference was noted between diabetic patients and non-diabetic ones in terms of OPG (16.78 ± 3.37 vs. 8.50 ± 3.88 pmol/L, respectively, *p* < 0.001), as well as in CaSc (424 (IQR, 144 to 928) vs. 118 (IQR 41 to 533), respectively, *p* < 0.001). Multiple correlations between variables adjusted by diabetes are shown in Table 2; OPG correlated positively with age, cCa, and VC, and negatively with albumin. With respect to treatments, we did not find differences between dialysis modalities, CAPD, or APD, in regard to OPG levels.

### 3.2. Follow-up

During a two-year follow-up, 34 patients died. Causes of death were: CVD in 15 patients (44%), including acute myocardial infarction, sudden death, arrhythmia, and heart failure, PD-related peritonitis in 2 patients (5.9%), non-peritoneal-related infections in 2 patients (5.9%), uremia/hyperkalemia in 5 patients (14.7%), cancer/stroke in 2 patients (5.9%), hypovolemic shock in 2 patients (5.9%), and hyperglycemia/acidosis in 6 patients (17.7%).

Table 3 shows the differences in patient demographics, as well as clinical and biochemical baseline data between the survivors and non-survivors. Non-survivors were older, more frequently diabetic, and had higher values of systolic blood pressure, OPG, and a higher incidence and severity of vascular calcifications. They also had lower levels of serum albumin.

Table 4 presents the results of the unadjusted Cox model analysis (univariate analysis) of factors associated with all-cause and CV mortality. The presence of diabetes mellitus increases by 82% and 85%, respectively, the possibility of having the risk of all-cause and CV mortality. Diabetes mellitus, age (by year), systolic blood pressure (for each mmHg), OPG (pmol/L), VC (for each CaSc), and low albumin (for each g/dL) were predictors for all-cause mortality while, diabetes, age, OPG, cCa, VC, and decreased Alb were predictors for CV mortality.

Table 5 shows the multivariable analysis (adjusted Cox analysis) of mortality with baseline data of calcium metabolism biomarkers (Model 1); high OPG and low albumin levels were associated with all-cause mortality. OPG, cCa, and iPTH were associated with CV mortality. OPG was significant in both mortalities. Model 2 adds DM to Model 1. OPG was significant only in all-cause mortality; cCa and PTH were significant in cardiovascular mortality. Model 3 adds age to Model 2. Albumin was significant for all-cause mortality. cCa, PTH, and age were significant for CV mortality. OPG was not significant. This was due to the high association and collinearity that OPG had with age, as was shown in multiple correlations.

Vascular calcification was not a risk factor in multivariate analysis of either all-cause or CV mortality.

To know the categorical value of each variable in combination with all other variables, we performed Cox regression analysis using the forward conditional method (Table 6). The highest score, according to all-cause mortality, was obtained by OPG (18.77, *p* < 0.001, followed by sAlb (13.84, *p* < 0.001), DM (13.67, *p* < 0.001), Age (10.56, *p* > 0.006), VC (5.59, *p* > 0.018), and SBP (5.24, *p* < 0.022). For CV mortality; OPG had the highest score (11.90, *p* < 0.001), followed by cCa (11.07, *p* <0.001), VC (10.38, *p* < 0.001), DM (8.70, *p* < 0.003), Age (7.55, *p* < 0.006), and Alb (5.99, *p* < 0.014). With this analysis, it was possible to confirm the highest value of OPG as a predictor of death in combination with markers of calcium metabolism, inflammation, DM, and age.

Figure 2a shows the diagnostic value of OPG for all-cause mortality in a ROC curve, where the area under the curve (AUC) was 0.72, *p* < 0.001 (95% CI: 0.627–0.821). The Youden index was identified as 14.37 pmol/L as the OPG cut-off value, with a sensitivity of 72.4% and a specificity of 62.5%. Figure 2b shows the diagnostic value of OPG for CV mortality in a ROC curve, where AUC was 0.70, *p* < 0.011 (95% CI: 0.552–0.845). The Youden index was identified as 13.57 pmol/L as the OPG cut-off value with a sensitivity of 77.3% and a specificity of 64.8%.

Figure 3a shows the association of OPG concentration with all-cause mortality (Cox analysis), classified as 14.37 pmol/L. It is important to note that the highest OPG concentration (>14.37 pmol/L) had the lowest survival compared to those of the lowest OPG levels (<14.37 pmol/L), with an HR of 0.203 (95% CI: 0.096–0.43, *p* < 0.001). Figure 3b shows the association of OPG concentration with cardiovascular mortality (Cox analysis), classified as 13.57 pmol/L. It is important to note that the highest OPG concentration (>13.57 pmol/L) had the lowest survival compared to those of the lowest OPG levels (<13.57 pmol/L), HR of 0.198 (95%CI 0.006–0.432, *p* < 0.006).

## 4. Discussion

Data from this study suggest that high OPG concentrations, a molecule related to mineral metabolism, inflammation, and vascular calcification, have a high predictive value for both all-cause (>14.37 pmol/L) and cardiovascular mortality (>13.57 pmol/L) in incident PD patients. Its predictive value is greater than those of other commonly used biomarkers, such as VC. Low albumin and high cCa levels were risk factors for only all-cause and cardiovascular death.

The novelty of this study resides in the fact that the PD cohort was composed of relatively young patients and that the selection criteria excluded those with higher mortality risk. The adopted inclusion/exclusion criteria allowed us to verify whether OPG, already associated with mortality in older cohorts of dialysis patients, was a mortality predictor in younger ones, a population in which focusing on cardiovascular risk may be of practical relevance.

The cut-off point values of OPG >14.37 pmol/L and >13.57 pmol/L were defined as a higher risk factor for all-cause and CV death using ROC curves. This was one of the main findings of the present study and is not surprising because of its relevance as a risk factor for the development and progression of heart valve calcification in PD patients [21], thoracic and femoral arterial calcification [22], as well as atherosclerosis, all-cause mortality, and cardiovascular dysfunction, which have already been demonstrated in other studies in hemodialysis and 3–5 CKD patients [23,24,25,26]. A recent meta-analysis showed that elevated circulating OPG levels independently predicted an increased risk for CV mortality in patients with CKD [27].

In the in vitro studies, OPG inhibits vascular calcification and protects endothelial cells from apoptosis, and it also promotes neovascularization in vivo as it is a soluble decoy receptor for TRAIL and RANKL [28,29,30,31]. In kidney patients, serum OPG levels increase, which is associated with vascular calcification and cardiovascular disease [11]. This apparent paradox can be understood as a compensatory mechanism to counteract calcification, endothelial damage, and ongoing inflammation. In this way, the positive association between OPG and diabetes, age, systolic blood pressure, calcium, vascular calcifications, as well as the negative correlation with serum albumin, found in the present study, can be explained.

Mechanisms other than mineral metabolism may be involved in increasing OPG, including the expansion of extracellular volume. In healthy humans, high sodium intake significantly elevates OPG in the blood. On the other hand, chronic inflammation through the increase in pro-inflammatory cytokines can also be a stimulus for an increase in circulating OPG [32]. Both conditions, extracellular expansion, and chronic inflammation, are frequent findings in PD patients [33]. Some studies have suggested that high levels of circulating OPG and inflammation have independent and additive values as predictors of death in patients with CKD and end-stage renal disease (ESRD) [34]. The lack of association of OPG with hsCRP in this and other studies does not invalidate the association, since hsCRP is not the only marker of inflammation [26]. As a member of the TNF superfamily, OPG may be involved in several inflammatory pathways. It is also important to mention that the OPG–inflammation relationship is bidirectional, and OPG can regulate the expression of interleukins in response to inflammatory stimuli, while its expression and production are regulated by several cytokines [35].

The ability of pro-inflammatory mediators, such as TNFα, interleukin-1, and platelet-derived growth factor, to enhance OPG expression and production in vascular cells, may explain the association between OPG concentrations and cardiovascular diseases [36].

Another important finding of our results was that OPG was a better predictor of both cardiovascular and all-cause mortality than VC or other biomarkers of calcium metabolism and inflammation. This suggests that OPG is associated with vascular damage and mortality through mechanisms independent of VC.

In keeping with our results, the authors of a published review concluded that circulating OPG levels could be used as an independent biomarker of cardiovascular disease in patients with acute or chronic cardio-metabolic diseases to improve the prognosis [37].

The useful prognostic value of calcification in patients with CKD is doubtful. It has been shown, in previous studies, that the quantification of coronary artery calcifications (CAC) with CT is valid for patients without CKD; however, VC association with cardiovascular death is not significant in CKD patients when adjusted for cardiovascular risk factor markers [38]. Our study confirms the findings that vascular calcifications in multivariate analysis (in combination with biomarkers of calcium metabolism and inflammation) were not associated with the risk of all-cause and cardiovascular death.

Although cardiovascular risk stratification with vascular calcification is used in the clinical setting, it is rather impractical for routine use. On the contrary, OPG, albumin, and cCa level determinations are widely available and can be repeatedly tested during patient follow-up by means of routine laboratory assessments.

Previous studies in PD patients showed that cCa was significantly associated with cardiovascular mortality and the results of our study are in line with these findings [39,40].

Serum albumin (sAlb) was the other circulating marker found to be associated with all-cause death. Previously, in several clinical studies, sAlb has been shown to be associated with CV death. It is speculated that hypoalbuminemia is a surrogate marker of proteinuria or inflammation [41].

The practical implications of the present study in the care of PD patients are that OPG and cCa levels could be an alternative or a complementary evaluation in the assessment of cardiovascular risk in PD patients. VC assessment, by means of a CT scan, could be reserved for specific cases due to its limited availability, high cost (150 USD compared to 13 USD for an OPG test) and its major operator dependency.

This study has some limitations; most importantly, the sample size is possibly considered to be too small. However, selection criteria allowed us to analyze a representative sample of a common population of PD patients in Mexico. Moreover, our study is the first, to date, to include only incident patients on PD and determine the risk of cardiovascular death, taking VC, OPG, and biochemical markers of mineral metabolism into account.

## 5. Conclusions

The data presented here show that OPG concentrations above 14.37 and 13.57 pmol/L have the highest all-cause and cardiovascular mortality predictive values in incident PD patients. The OPG predictive value overcomes other biomarkers, such as vascular calcification. Its systematic use could help in identifying patients with higher mortality risk, with the aim of providing a more intensive follow-up and adapting treatment. Future research could be conducted in intervention studies.

## Data Availability

The database used in the current study is not available in a public repository, but they are available from the corresponding author on reasonable request.

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
