# Peer review of "Osteoprotegerin Is a Better Predictor for Cardiovascular and All-Cause Mortality than Vascular Calcifications in a Multicenter Cohort of Patients on Peritoneal Dialysis"

_biomolecules, 2022, doi:10.3390/biom12040551_

Round 1

Reviewer 1 Report

Biomolecules; biomolecules-1637632

Title: Osteoprotegerin, is better predictor for cardiovascular and all-cause mortality than vascular calcifications in a multicenter cohort of patients on peritoneal dialysis

Avila and co-authors conducted a longitudinal study on the predictive value of vascular calcification and serum osteoprotegerin for cardiovascular mortality in patients on peritoneal dialysis. In brief, data (biochemical assays, multi-slice computed tomography, Cox regression analyses) were extracted from 197 patients among seven dialysis centers in Mexico City. As a result of their findings, the authors conclude that osteoprotegerin levels (vs. vascular calcification assessments) have the highest predictive value as a risk factor for all-cause and cardiovascular death in dialysis patients. See comments below.

(1) If possible, it would be desirable to see some sample tomography images and visual illustration(s) as to how data were extracted for vascular calcification indices.

(2) Where reasonable and appropriate, the authors should indicate numbers/data to precisely define their use of “high” or “low” levels of various biomarkers throughout the manuscript (starting with the Abstract).

(3) For Figure 2, both y-axes for 2a & 2b and the legend should indicate “sensitivity” instead of “sensibility”. The y-axis of Figure 3B should indicate “cumulative” and not “cumulated”.

(4) Multiple different font styles have been used across text and Figures throughout the manuscript. 

Reviewer 2 Report

This study aimed at evaluating the predictive role of steoprotegerin and other biomarkers (serum albumin, albumin-corrected calcium) of all-cause and Cardiovascular mortality in patients on Peritoneal Dialysis in comparison with vascular calcification. The topic is of interest, and the results are generally clearly described and sufficiently discussed, although some points needs to be clarified. Furthermore, language requires extensive revision by a native speaker due to the presence of several grammar errores, and a list of abbreviations should be provided. Specific comments are listed below.

Abstract

Third line. Please replace “recuitment” with “recuited”.

Consider changing “Cox analysis” to “Cox regression model”. Besides, replace “was” with “were”.

In the sentence “The Median OPG was 11.28; (IQR: 7.6-17.4 pmol/L…” please add the round bracket after “pmol/L”.

Introduction

Please add the full name of “HD”

At the end of introduction provide the acronym for Peritoneal Dialysis.

Study design

Consider deleting “with” after “7 hospitals”.

Patient selection

The last sentence “Informed consent was obtained from all individual participants included in the study” was already written at the end of the first paragraph of this sub-section.

Statistical analysis

The full names of biomarkers were already reported in “Biochemical assessments”.

Please delete the point before “With significance p<0.05”.

Results

Figure 1. In the flow chart please correct the typo in “assessed”.

What is CaSC? It is unclear, and this acronym was not reported in Table 2.

Table 3. Among the survivors, there is a lower proportion of diabetic patients, in contrast to what the authors said.

Table 6. The second highest score for all-cause mortality is not that of cCA but that of sAlb.

Discussion

In the first sentence, please delete the semicolon after “both”.

“In experimental studies, OPG inhibits vascular calcifications, protects endothelial cell from apoptosis and promotes neovascularization in vivo”. How can you explain this finding, being OPG associated with vascular calcification and cardio-vascular disease in human studies?

“…both characteristic of PD patients”. Please change “characteristic” with “characteristics”.

The paragraph “Nevertheless, some studies suggested that high circulating OPG levels and inflammation have independent, additive value as predictors of death in patients with CKD and ESRD [33]. Despite the lack of association with hsCRP in the pre-sent study, as similar with other study [26] . OPG could be involved in inflammatory path-ways as a member of TNF superfamily. Indeed, it is able to regulate the expression of interleukins in response to inflammatory stimuli” is confusing. Please check the grammar and rephrase. Furthermore, the full name of ESRD is missing.

“Another important finding of our results were that OPG”. Please replace “were” with “was”.

Reviewer 3 Report

In the present prospective observational study, the authors evaluated the predictive value of serum osteoprotogerin (OPG) levels on cardiovascular (CV) and all-cause mortality in 197 incident patients on peritoneal dialysis at 7 centers in Mexico, and compared it with calcification vascular (VC), and with other biochemical markers. The authors concluded that serum levels of OPG have a high predictive value superior to CV, both in relation to CV mortality and all-cause mortality, and that they could serve as a useful and accessible instrument to assess CV risk in dialysis patients. The study is well written and the conclusions are supported by the results.

Author Response

Thanks for your comments